# Health-Related Quality of Life by Gulf War Illness Case Status

**DOI:** 10.3390/ijerph19084425

**Published:** 2022-04-07

**Authors:** Elizabeth J. Gifford, Stephen H. Boyle, Jacqueline Vahey, Kellie J. Sims, Jimmy T. Efird, Blair Chesnut, Crystal Stafford, Julie Upchurch, Christina D. Williams, Drew A. Helmer, Elizabeth R. Hauser

**Affiliations:** 1Cooperative Studies Program Epidemiology Center-Durham, Durham VA Medical Center, Durham VA Health Care System, Durham, NC 27705, USA; stephen.boyle@va.gov (S.H.B.); jvahey@alum.mit.edu (J.V.); jimmy.efird@va.gov (J.T.E.); blair.chesnut@duke.edu (B.C.); crystal.stafford@va.gov (C.S.); julie.upchurch@va.gov (J.U.); christina.williams4@va.gov (C.D.W.); elizabeth.hauser@duke.edu (E.R.H.); 2Center for Child and Family Policy, Duke Margolis Center for Health Policy, Duke University Sanford School of Public Policy, Durham, NC 27708, USA; 3Computational Biology and Bioinformatics Program, Duke University School of Medicine, Durham, NC 27705, USA; 4Department of Radiation Oncology, School of Medicine, Case Western Reserve University, Cleveland, OH 44106, USA; 5Center for Innovations in Quality, Effectiveness and Safety (IQuESt), Michael E. DeBakey VA Medical Center, Houston, TX 77030, USA; drew.helmer@va.gov; 6Department of Medicine, Baylor College of Medicine, Houston, TX 77030, USA; 7Department of Biostatistics and Bioinformatics, Duke University Medical Center, Duke Molecular Physiology Institute, Durham, NC 27710, USA

**Keywords:** Gulf War illness, health-related quality of life, post-traumatic stress disorder, sleep disturbance, pain, depression

## Abstract

This study examines how health-related quality of life (HRQOL) and related indices vary by Gulf War illness (GWI) case status. The study population included veterans from the Gulf War Era Cohort and Biorepository (*n* = 1116). Outcomes were physical and mental health from the Veterans RAND 12 and depression, post-traumatic stress (PTSD), sleep disturbance, and pain. Kansas (KS) and Centers for Disease Control and Prevention (CDC) GWI definitions were used. Kansas GWI derived subtypes included GWI (met symptom criteria; no exclusionary conditions (KS GWI: Sym+/Dx−)) and those without GWI: KS noncase (1): Sym+/Dx+, KS noncase (2): Sym−/Dx+, and noncase (3): Sym−/Dx−. CDC-derived subtypes included CDC GWI severe, CDC GWI mild-to-moderate and CDC noncases. Case status and outcomes were examined using multivariable regression adjusted for sociodemographic and military-related characteristics. Logistic regression analysis was used to examine associations between GWI case status and binary measures for depression, PTSD, and severe pain. The KS GWI: Sym+/Dx− and KS noncase (1): Sym+/Dx+ groups had worse mental and physical HRQOL outcomes than veterans in the KS noncase (2): Sym−/Dx+ and KS noncase (3): Sym−/Dx− groups (*p*s < 0.001). Individuals who met the CDC GWI severe criteria had worse mental and physical HRQOL outcomes than those meeting the CDC GWI mild-to-moderate or CDC noncases (*p*s < 0.001). For other outcomes, results followed a similar pattern. Relative to the less symptomatic comparison subtypes, veterans who met the Kansas symptom criteria, regardless of exclusionary conditions, and those who met the CDC GWI severe criteria experienced lower HRQOL and higher rates of depression, PTSD, and severe pain.

## 1. Introduction

Following deployment in support of the 1990–1991 Persian Gulf War, an estimated quarter to a third of veterans returned with what is now known as Gulf War illness (GWI) [1,2]. GWI has been associated with an enduring presence of symptoms such as fatigue, pain, and cognitive or mood symptoms that may interfere with veterans’ physical and mental health, referred to here as health-related quality of life (HRQOL). Decreased physical and mental HRQOL have been documented among veterans deployed to the Persian Gulf relative to non-deployed Gulf War era veterans from the U.S.A., Australia, and the United Kingdom [3,4,5,6,7,8,9,10,11,12,13]; these associations were identified shortly after the war [3] and have persisted for decades [6]. Yet, there is heterogeneity in how GWI presents, including the combination and severity of symptoms. Thus, the current study seeks to understand how GWI is associated with veterans’ HRQOL and related health indices by comparing Gulf War era veterans across subtypes of GWI case status as assessed approximately 25 years following the war.

Self-reported HRQOL is an important construct for understanding veterans’ concerns as they seek health care [14,15,16,17]. Studies evaluating HRQOL commonly used variations in instruments originally developed for the Medical Outcomes Study, including the 36-item Short Form Health Survey (SF-36), the abbreviated 12-item measure (SF-12), and versions adapted for veteran populations (e.g., Veterans RAND 12) [15,18,19]. Importantly, these related measures assess how the respondents’ mental and physical health affects their physical and social functioning, and their ability to complete work or other daily activities [15,18,19]. Among veterans, lower HRQOL has been associated with higher rates of hospitalizations, primary care visits, specialty medical visits, limitations in activities of daily living, and even death [20,21].

We identified six studies that compared composite mental and physical health measures of HRQOL among Gulf War era veterans with and without GWI [1,22,23,24,25,26]. The earliest study, conducted by Fukuda and colleagues [1], established the base CDC GWI definition and provided a severe GWI definition, which only considered symptoms that were reported as severe. Results revealed that individuals classified as non-GWI cases (*n* = 59) had better mental and physical health scores than individuals classified as mild-to-moderate GWI cases (*n* = 86), who in turn had better scores than individuals classified as severe GWI cases (*n* = 13). Two studies analyzed samples from the 1995 National Health Survey of Gulf War Era Veterans and their Families (NHS) using follow-up information collected in 1999–2001 (*n* = 492 GWI cases and *n* = 1659 noncases) [22] and 2005–2008 (*n* = 111 noncases and 73 GWI cases) [26]. They found that mean mental and physical health scores were roughly a standard deviation lower among individuals who met the CDC GWI definition relative to noncases. Three case control studies similarly found that veterans who met GWI case criteria had worse physical and mental HRQOL scores than controls (number of noncases ranged from 16 to 52 while number of cases ranged from 25 to 223) [23,24,25].

HRQOL has been shown to be highly correlated with other indices of physical and mental health such as pain, depression, PTSD, and sleep disturbances [4,10,27,28,29,30]. For example, deployed relative to non-deployed Gulf War era veterans reported higher rates of chronic widespread pain, and those with chronic widespread pain reported lower quality of life than those without such pain [31]. Like pain, psychiatric morbidity (e.g., PTSD, depression, affective symptoms, anxiety symptoms) was related to poorer HRQOL after controlling for measures of physical health and social support [6]. Conceptually, Wilson and Cleary [17] describe how symptom states are dimensions of patients’ experiences that contribute to their broader understanding of factors that compose HRQOL.

The GWI case definitions recommended by the Institute of Medicine (IOM) for research [2] are the Kansas and the CDC. How GWI cases and control groups are distinguished by these definitions creates important sources of heterogeneity that complicates our understanding of the extent to which GWI is associated with veterans’ HRQOL. Both definitions rely on self-reported ratings of persistent (≥6 months) or re-occurring symptoms across multiple domains. With the Kansas GWI definition, veterans who meet the case-defining symptom criteria but have certain health conditions (e.g., diabetes, heart disease, and cancer) are considered noncases. Comparing HRQOL between noncases and cases likely underestimates the association between GWI and HRQOL because the comparison group contains two distinct subgroups: one that is comprised of relatively healthy veterans and another that is comprised of veterans with a significant condition that would be expected to negatively impact HRQOL. For the CDC GWI definition, some case defining symptoms, such as pain and difficulty getting to or staying asleep, are common in middle and older adulthood [32]; so much so that 80% of the non-deployed members of the Gulf War Era Cohort and Biorepository (GWECB) cohort (mean age at survey = 54.6 years, SD = 8.0) met the CDC GWI symptom criteria [33]. However, the CDC definition does also allow subtypes of GWI which are differentiated by the severity of case-defining symptoms. How exclusionary health conditions and GWI case-defining symptom severity are used to define GWI cases is an important consideration for understanding the association between GWI and HRQOL—especially as the cohort ages.

Our study aimed to extend previous work on the HRQOL of Gulf War era veterans by examining variation in HRQOL and related outcomes across subgroups of veterans based on GWI case status, as determined by two IOM-recommended definitions (i.e., Kansas and CDC) [34]. Our research question is “How do HRQOL and indices of health outcomes vary across subgroups of veterans based on GWI case status?” To answer this question, we used the GWECB data to examine self-reported information regarding HRQOL and other indices of health known to be more common among Gulf War veterans (i.e., scores of depression, PTSD, pain, and sleep disturbance) [35]. We further examined how subgroups of veterans based on GWI case status differed in rates of severe pain, clinical depression, and PTSD.

## 2. Materials and Methods

### 2.1. Study Population

This is a retrospective cross-sectional study. Participants were selected from the 1343 individuals who participated in the GWECB. Consistent with previous work [33], we sequentially excluded individuals who were missing deployment status (*n* = 21), Kansas GWI case status (*n* = 100), CDC GWI case status (*n* = 1), CDC severe case status (*n* = 13), sex (*n* = 17), and those who were deployed in support of the Gulf War but not to the theater of operations (*n* = 75), for a final sample of 1116 (see Appendix A for sample flow diagram).

The sampling and data collection procedures for the GWECB were previously reported [35] and are briefly described here. The GWECB sampled veterans who served in the US Uniformed Services between August 1990–July 1991. Eligibility was not restricted by deployment or combat status and importantly, the cohort included users and nonusers of the Veterans Health Administration (VHA) and oversampled women. The recruitment sample was drawn from a stratified random sample from the Defense Manpower Data Center (total population was nearly 5 million). Stratification factors included (1) deployment to the theater of operations or areas where combat-related activities occurred (50%); (2) military component (active duty 50%); (3) military branch (Army 50%); (4) grade of officer (officer 10%); (5) race/ethnicity (nonwhite race 20%), and (6) sex (nonmale sex including female and unknown 20%). The pilot recruitment sample included 10,042 individuals as well as 168 individuals who requested to participate. By design, the pilot sample matched the geographic distribution of the recruitment panel across US census regions. Cities within each census region served as a recruitment hub owing to the availability of phlebotomists.

### 2.2. Measures

#### 2.2.1. Gulf War Illness

This study drew on established case definitions of GWI—Kansas and CDC as previously operationalized for the GWECB cohort [33]. To meet the symptom criteria for the Kansas definition, a veteran needed to report multiple symptoms or report at least one symptom as moderate or severe in at least 3 of the 6 symptom domains: fatigue/sleep problems (4 symptoms); pain (3 symptoms); neurologic/cognitive/mood (14 symptoms); gastrointestinal (3 symptoms); respiratory (3 symptoms); skin (2 symptoms) (Appendix A lists symptoms in each domain). The symptom questions on the GWECB survey asked, “Over the past 6 months, have you had a persistent or reoccurring problem with…?” and for symptoms to which they responded positively, “How would you rate this problem?”—mild, moderate, or severe. To meet the Kansas GWI criteria, a veteran must have met the symptom criteria and reported no exclusionary conditions (KS GWI: Sym+/Dx−) (i.e., cancer, diabetes, heart disease, stroke, chronic infectious disease (HIV, tuberculosis, hepatitis C), liver disease, lupus, schizophrenia, bipolar disorder, multiple sclerosis, and traumatic brain disorder). Presence of the exclusionary conditions was determined based on self-reported responses to the GWECB survey question “Have you ever been told by a doctor or healthcare provider that you have…?”

Veterans not meeting the Kansas GWI case definition were categorized as one of three mutually exclusive groups: (1) those who met the symptom criteria and reported at least one exclusionary condition (KS noncase (1): Sym+/Dx+); (2) those who did not meet the symptom criteria and reported at least one exclusionary condition (KS noncase (2): Sym−/Dx+), and (3) those who met neither the symptom criteria nor the exclusionary criteria (KS noncase (3): Sym−/Dx−).

The CDC GWI case definition has three symptom domains—fatigue (1 symptom), musculoskeletal (3 symptoms), and mood-cognition (6 symptoms) (Appendix A lists each of the symptoms). Veterans met the CDC GWI criteria by having at least one symptom in at least two domains. The CDC GWI severe criteria require individuals to report at least one symptom as severe in at least two symptom domains. For this analysis, the CDC GWI case status is comprised of three mutually exclusive groups: CDC GWI severe, CDC GWI mild-to-moderate, and noncases.

#### 2.2.2. Health-Related Quality of Life (HRQOL)

HRQOL was measured using the Veterans RAND 12-item Health Survey (VR-12). The VR-12 was developed from the Veterans RAND 36-item Health Survey (VR-36), which was developed and modified from the original RAND version of the 36-item Health Survey version 1.0 (also known as the Medical Outcomes Survey (MOS) SF-36). It is comprised of 12 items that relate to 8 physical and mental health domains: general health perceptions, physical functioning, role limitations owing to physical problems, role limitations owing to emotional problems, bodily pain, energy/fatigue, social functioning, and mental health. Factor analytic studies have shown that the 12 items can be summarized as a physical component summary score (PCS-12) and a mental component summary score (MCS-12) [36]. These scores were derived using an algorithm developed from the 2000–2002 US Medical Expenditure Panel Survey population. The two component scales have been shown to explain over 90% of the variance in PCS and MCS scales of the VR-36 [37]. VR-12 scores are standardized as T-scores with a mean of 50 and a standard deviation of 10; higher scores on the two components indicate better physical and mental health. The VR-12 provides a reliable and valid measure of health status and has been widely used in studies of veterans’ health [38]. Scores were derived using an algorithm written in SAS statistical software, which includes code to impute missing values provided by the Center for the Assessment of Pharmaceutical Practices at Boston University School of Public Health.

#### 2.2.3. Sleep Disturbance

The 4-item Patient-Reported Outcomes Measurement System (PROMIS) sleep disturbance instrument was used to measure sleep quality [39]. The four items asked about sleep quality (i.e., “my sleep quality was…” rated from very poor to very good) and sleep problems (e.g., “I had difficulty falling asleep” rated from not at all to very much) on 5-point scales over the past 7 days. Responses were scored according to the criteria provided by the HealthMeasures scoring service [40], which allowed us to include participants who completed at least three of the four items. This scale was used to calculate a T-score (mean = 50, SD = 10), with higher scores reflecting greater sleep disturbance. GWECB participants’ responses to the four PROMIS sleep disturbance questions were then matched to the corresponding score calculated by the HealthMeasures scoring system.

#### 2.2.4. Post-Traumatic Stress Disorder (PTSD)

PTSD was measured using the 17-item Post-traumatic Checklist Civilian version (PCL-C). The items reflect the DSM-IV PTSD symptom criteria. Respondents indicated how bothered they had been by each symptom over the past month using a 5-point scale (not at all (1)–extremely (5)). Scores range from 17–85 and the internal consistency of the scale has been reported as 0.96 [41]. The 17 questions of the PCL-C can be divided into three clusters: re-experiencing (5 items), avoidance/numbing (7 items), and hyperarousal (5 items). To calculate a total score, up to 1 item per cluster could be missing. When a single item within a cluster was missing, the average of the other items in the cluster were used to impute the score of the missing item. The total score was calculated as the sum of the scores of the three clusters. Although there is not an exact cutoff point to identify whether or not someone suffers with PTSD, the National Center for PTSD offers guidelines for the PCL-C use for specific populations, including cutoff scores for 45–50 for VA or civilian specialty mental health clinics [42]. We selected 50 as the cutoff point for this sample to lessen the chance of false positives in our sample of GW veterans, a population with relatively high rates of documented PTSD [43].

#### 2.2.5. Depressive Symptoms and Depression

Severity of depressive symptoms was assessed by the Patient Health Questionnaire 8-item depression scale (PHQ-8) [44]. The item content of the PHQ-8 aligns with the Diagnostic and Statistical Manual of Mental Disorders, fourth edition (DSM-IV) criteria for depression. This scale did not include the item assessing suicidal/self-injurious intention which is deemed inappropriate in a community sample where the researcher is unable to connect an individual to resources. Respondents were asked how often in the past 2 weeks they had been bothered by a particular symptom of depression on a 4-point scale (not at all (0)–nearly every day (3)) with total scores ranging from 0–24 and an internal reliability estimated as high (Cronbach’s α = 0.91). Total scores were not calculated for individuals who had more than 1 item missing. For individuals who completed 7 of the 8 items of the PHQ-8, the missing item was imputed with the average of the 7 completed questions. To assess depression, we constructed a binary variable indicating whether the PHQ-8 total scores were greater than or equal to 10. A PHQ-8 score of ≥10 has been shown to have a high sensitivity and specificity for major depression [44].

#### 2.2.6. Pain

Pain was assessed with one question on an 11-point scale (0–10): “On a scale of 0–10, where 0 means no pain and 10 means pain as bad as you can imagine, please rate your overall mount of pain in the PAST WEEK” To assess clinically relevant pain, we created a binary variable indicating whether or not the veteran reported a score of 7 or greater [45].

#### 2.2.7. Demographic and Military Characteristics

Other measures are described in more detail elsewhere [33] and included: deployed to the Gulf vs. not deployed in support of the Gulf War, age at time of survey (continuous), sex (male, female, unknown/other), race/ethnicity (White non-Hispanic, Black non-Hispanic, Hispanic Other), income (<$30,000, $30,000–$59,999, $60,000–$99,999; $10,000+, unknown), education (<Associates degree, Bachelors or Associates degree, Masters, Professional, or Doctorate degree), military component (active duty only, active duty and reserves, reserves only), and military branch (Army, Navy, Airforce, Marines, National Guard, and other).

### 2.3. Analysis

Chi-squared, *t*-tests, and tests of proportions were used to examine bivariable relationships and assess whether the GWI measures (Kansas and CDC) varied by key demographic and military service-related characteristics. Pairwise Pearson Correlations were conducted to explore the extent to which outcomes were interrelated. For the Kansas definition, we were specifically interested in documenting differences between the group of veterans who met the KS GWI: Sym+/Dx−) with the subgroups of veterans who did not meet the Kansas GWI criteria (KS noncase (1): Sym+/Dx+, KS noncase (2): Sym−/Dx+, and KS noncase (3): Sym−/Dx−). In general, we hypothesized that veterans who met the Kansas GWI criteria would fare worse on measures of HRQOL and indices of health than the KS noncase (2): Sym−/Dx+ and KS noncase (3): Sym−/Dx− groups. In contrast, given the potential physical and mental health impairments potentially associated with the exclusionary conditions, we hypothesized that veterans who meet the Kansas GWI criteria would fare better on measures of HRQOL and indices of health than the KS noncase (1): Sym+/Dx+. For the CDC definition, we were specifically interested in differences between the CDC GWI severe and (1) CDC GWI mild-to-moderate and (2) CDC noncases. We hypothesized that veterans with CDC GWI severe would fare worse on measures of HRQOL and indices of health than veterans with CDC GWI mild-to-moderate, and in turn, those with CDC GWI mild-to-moderate would fare worse than those who were classified as CDC noncases.

Multivariable multiple regression was used to assess the association between GWI (Kansas and CDC) case status and HRQOL (physical (PCS-12) and mental (MCS-12) component summary scores), and total scores of sleep disturbance, PTSD symptoms, depressive symptoms, and pain. Models were estimated separately for the Kansas and CDC GWI definitions. As indicated previously, Kansas GWI status was modeled as a four-category variable with the KS GWI: Sym+/Dx− group serving as the reference group, and CDC GWI case status was modeled as a three-category variable with CDC GWI severe serving as the reference group. Models were adjusted for characteristics such as sex, age, race/ethnicity, income, and education, which previously have been associated with HRQOL measures [34,46,47] and military characteristics (deployment status, military unit, and military branch) associated with GWI [33,48]. Post-estimation linear combination tests were used to test differences in the size of the coefficients across GWI subtypes. Predicted means by GWI case status were generated from regressions for display purposes.

To assess the association of GWI case status and clinically relevant levels of depression (PHQ8 ≥ 10), PTSD (PCL-C ≥ 50), and severe pain (pain ≥ 7), logistic regression models were estimated and adjusted for age, sex, race/ethnicity, education, income, deployment status, military component, and military branch. Post-estimation linear combination tests were used to test differences in the size of the coefficients across GWI subtypes. Adjusted proportions at the means of other covariates were computed using the margins command in Stata for display purposes [49]. The criterion for judging statistical significance was *p* < 0.05. Analyses were performed using Stata version 16 and SAS statistical software version 9.4 (SAS Institute Inc., Cary, NC, USA) [49].

## 3. Results

### 3.1. Sample Description

Veterans participating in the GWECB cohort were predominantly male (76.8%), white (65.1%), and <60 years of age (61.3%). In terms of socioeconomic status, most veterans in the cohort reported a household income greater than or equal to $60,000 (58.7%) and having at least an Associate’s degree (91%). Regarding military characteristics, the veterans were predominantly on active duty only (60.8%) and deployed to the Gulf (76.1%), and about half were in the Army (45.5%). Finally, about half of the veterans were a VHA user (44.6%). In the GWECB cohort, 38.2% met the KS GWI: Sym+/Dx−, while 30.9% were KS noncase (1) Sym+/Dx+, 9.5% were KS noncase (2): Sym−/Dx+, and 21.4% KS noncase (3): Sym−/Dx−. For the CDC GWI definition, 23.7% met the CDC GWI severe case criteria, 59.6% met the CDC GWI mild-to-moderate, and 16.8% were CDC noncases. Table 1 presents the results of tests of proportions comparing demographic and military characteristics between groups using KS GWI- (and alternatively CDC severe GWI) as the reference. Appendix A displays a crosstabulation of Gulf War illness case status across the two definitions.

### 3.2. Correlations among HRQOL and Indices of Health

All the mental and physical health variables were significantly intercorrelated (see Appendix A). The correlation between the physical and mental component summary scores of the VR-12 was 0.29, *p* < 0.001 supporting the notion that these scales measure mostly independent dimensions of HRQOL. The strongest correlate of the physical component summary score of the VR-12 was the pain scale (r = −0.70, *p* < 0.001) and the strongest correlates of mental component summary score of the VR-12 were measures of depression (−0.81, *p* < 0.001) and PTSD (−0.77, *p* < 0.001).

### 3.3. Association of Gulf War Illness Case Status with Health-Related Quality of Life, Sleep Disturbance, PTSD, Depression, and Pain Measures

Unadjusted means of our measures of HRQOL by Gulf War case status are presented in Table 2. When considering the Kansas definition, the KS GWI: Sym+/Dx− group had better mean HRQOL scores (*p* < 0.001) on all measures (i.e., physical component summary score, mental component summary score, PTSD, depression, and pain) than those with KS noncase (1): Sym+/Dx+. However, the two groups did not differ on mean sleep disturbance (*p* = 0.22). In contrast, the KS noncase (2): Sym−/Dx+ and KS noncase (3): Sym−/Dx− groups had better mean functioning scores (*p* < 0.001) on all measures than those who met the GWI symptom criteria (Sym+). When using the CDC definition, individuals classified as CDC GWI severe had consistently worse mean HRQOL scores than either those classified as CDC GWI mild-to-moderate or CDC noncases.

The pattern of results observed in the unadjusted means remained consistent after adjusting for age (continuous), sex, race/ethnicity, education, income, deployment status, military component, and military branch (Figure 1, Appendix A). For CDC, those who met the CDC GWI severe criteria scored significantly worse on all measures than those who met the CDC GWI mild-to-moderate criteria, and in turn, those who met the CDC GWI mild-to-moderate criteria scored significantly worse than CDC noncases. Supplement to the regression results in Appendix A, contrasts between groups within each GWI definition are presented in Appendix A. A pattern of results that emerged was that the KS noncase (1): Sym+/Dx+ group had worse outcomes than the KS GWI: Sym+/Dx− group on all outcomes examined here (*p* < 0.001).

Finally, we present the unadjusted (see Appendix A) and adjusted proportion of clinically meaningful levels of PTSD (PCL-C ≥ 50), depression (PHQ8 ≥ 10), and severe pain (pain ≥ 7) (Figure 2). The unadjusted proportions were not substantively different from the adjusted proportions (Appendix A); thus, we focus here on our description on the adjusted means. As seen in Figure 2 and Appendix A, the adjusted proportion of individuals who met the clinical cutoff for PTSD was 18.5% (95% CI: 14.4%, 22.6%) among the KS GWI: Sym+/Dx− criteria and 32.3% (95% CI: 26.6%, 37.9%) for the KS noncase (1): Sym+/Dx+ group. This pattern was similar for depression where 28.8% (95% CI: 24.1%, 33.5%) of those who met the KS GWI: Sym+/Dx− criteria and 46.9% (95% CI: 40.9% 52.9%) of those in the KS noncase (1): Sym+/Dx+ group met the criteria for depression. For severe pain, the rates were 21.6% (95% CI: 17.3%**,** 25.8%) for the KS GWI: Sym+/Dx− group relative to 33.4% (95% CI: 27.8%, 39.0%) for the KS noncase (1): Sym+/Dx+ group. In using the CDC GWI criteria, a similar pattern emerged. Relatively few of the CDC noncases met cutoff values for PTSD, depression, and severe pain (all proportions <6%). In contrast, among those who met the CDC GWI severe criteria, 54.9% (95% CI: 47.4%, 62.5%) met the PTSD threshold, 76.4% (95% CI: 70.4%, 82.4%) met the depression threshold, and 53.0% (95% CI: 47.4%, 62.5%) met the severe pain threshold. The proportions in the CDC GWI severe group were 4- to 6-fold higher than GWECB participants in the CDC GWI mild-to-moderate group.

## 4. Discussion

Decades following the Gulf War, our results indicate that a significant proportion of veterans in our cohort suffer from a constellation of symptoms that signify GWI, and in turn, are associated with worse physical and mental health, and higher levels of sleep disturbance, PTSD, depression, and pain. Findings supported the hypothesis that both groups of veterans who met the Kansas GWI symptom criteria fared worse on these outcomes than veterans in the KS noncase (2): Sym−/Dx+ and KS noncase (3): Sym−/Dx− groups. Importantly, the KS noncase (1): Sym+/Dx+ group fared worse than the KS GWI: Sym+/Dx− group on most outcomes, with the only exception being sleep disturbance where these two groups did not differ. Further support for our hypotheses was seen in that veterans who met the CDC GWI severe category had worse scores on all outcomes studied here while those who met CDC GWI mild-to-moderate fared worse than veterans who did not meet CDC GWI on all outcomes.

Our study affirmed findings of previous studies and extended that work by considering multiple definitions of GWI (Kansas and CDC) and subclassifications based on these two definitions. Our findings demonstrated relatively lower HRQOL outcomes for veterans who, despite meeting the Kansas GWI symptom criteria, did not meet the Kansas GWI case definition due to exclusionary health conditions. These findings were not unexpected because many of the exclusionary conditions have been associated with poorer physical and mental health in studies of other populations [50,51]. When the Kansas definition was developed, the exclusionary health conditions did not differ by deployment status for Gulf War era veterans [48], a finding also observed in this cohort [33]. However, a recent study found that relative to non-deployed Gulf War era veterans, those deployed to the Gulf War had higher rates of certain health conditions considered exclusionary—such as coronary heart disease, as well as other conditions such as irritable bowel syndrome, gastritis, and dermatitis [52]. To both account for aging among Gulf War veterans and to further our understanding of how GWI may be associated with aging-related health conditions, our findings suggest the need to re-evaluate how exclusionary health conditions are incorporated in the determination of GWI case status when considering the Kansas definition. This re-evaluation is important for ensuring individuals with comorbidities are not falsely excluded from classification as a GWI case. Further, this line of inquiry is critical for allowing research that examines whether or not GWI is associated with the occurrence of health conditions.

Regardless of deployment status, four out of five veterans in our cohort met the CDC GWI criteria [33]. Thus, we focused on the CDC GWI severe category which restricted case-defining symptoms as those that the veteran reported as severe. Striking differences, on the order of 4 to 6.5 times, in the rates of depression, PTSD, and severe pain were observed in the CDC GWI severe group vs. the CDC GWI mild-to-moderate group. This calls for incorporating symptom severity into research studies and clinical practices for veterans with GWI.

Veterans with CDC GWI severe and Kansas GWI symptoms (both with and without exclusionary criteria) were among the groups in our cohort with the highest rates of depression, PTSD, and severe pain. Depression and PTSD have been prospectively associated with exclusionary conditions, such as cardiovascular disease [53,54] and diabetes [55,56]. Pain has also been associated with mental health outcomes including depression [57] and PTSD [58]. Whereas it is believed that psychological distress can enhance the severity of GWI symptoms [59], it is also possible that the chemical exposures during the Gulf War could negatively affect cognition and mood through enhanced oxidative stress and inflammatory activity, as well as through alterations in the gut microbiome [60]. In our cross-sectional study, it is not possible to examine the temporal ordering of GWI case status and medical and psychiatric comorbidities.

One challenge in GWI research has been applying our current definitions of GWI, which were constructed based on data gathered shortly after service members returned from the Gulf War, to data collected decades following the war. The exclusionary health conditions used in the Kansas definition and the symptoms used in both definitions tend to become more prevalent with age. Thus, applications of the Kansas definition should carefully consider how to incorporate study participants who meet the symptom criteria for GWI but who also have an exclusionary condition. Similarly, a substantially larger proportion of the GWECB participants who met the CDC GWI severe criteria had worse outcomes. Future research should consider if use of the CDC GWI severe criteria may be warranted and if this is too restrictive of a criterion.

While Gulf War era veterans who meet the criteria for GWI have indicators of poorer physical and mental health, some of the comorbidities experienced by these veterans, including sleep disturbance, pain, depression, and PTSD, have been the target of treatments effective in alleviating some patients’ distress. In fact, a recent systematic literature review of studies regarding the effectiveness of treatments for individuals with GWI [61] revealed that mindfulness-based stress reduction improved several outcomes related to these domains, including pain, fatigue, depression, and PTSD. Thus, further understanding the extent to which alleviating related symptoms might improve aspects of HRQOL for veterans with GWI is warranted.

### Strengths and Limitations

Findings of this study have implications across a variety of constituents. For the scientific community, this is one of the few studies that has examined HRQOL and indices of health across subtypes of GWI. Future work that seeks to examine whether and how GWI is related to the aging process may benefit from carefully considering the implications of how diagnosable health conditions and severity of symptoms are incorporated. For veterans, their social networks, and clinicians caring for the needs of veterans with GWI, an understanding of the lower physical and mental HRQOL, and the higher frequencies of sleep disturbance, PTSD, depression, and pain may help to document the health burden that veterans with GWI face. It may be possible to improve some aspects of veterans’ lives by focusing on interventions and treatments targeted at these health concerns. By documenting the health burdens faced by veterans, findings from this study may help to showcase the persistent needs of veterans with GWI.

Despite efforts to obtain a representative sample of Gulf War era veterans through stratified random sampling, the final response rate was low (12.5%) and certain groups (e.g., non-deployed veterans) were underrepresented in the final sample [35]. This may limit the generalizability of findings observed in this cohort to the broader population. Self-reported symptoms may be subject to biases shared with other self-reported health status measures; thus, it is important to examine associations between GWI and health problems verified from other sources, such as a physician’s diagnosis. Our approach was to accept veterans’ self-reports as valid indicators of their symptoms regardless of reported mental health conditions. While those experiencing cognitive burdens may have had trouble completing the survey, this likely would have led to non-response or inconsistent symptom reporting and would not have produced the strong pattern of results seen here. We did not have information on whether or not the veterans attributed their symptoms to the Gulf War, which other studies have found to be associated with the extent to which veterans are bothered by their symptoms and worse clinical outcomes [59,62]. This study examined cross-sectional associations between GWI and HRQOL and related measures of health, so we cannot determine causality or directionality of effects. Thus, we did not examine the extent to which GWI was associated with the various mental and physical HRQOL indices above and beyond the association of each subtype with each individual measure (e.g., depression). Furthermore, additional factors may affect the outcomes studied here such as physical and cognitive disabilities. The constructs used here to assess HRQOL, indices of health, and GWI share common elements. Still, the validated constructs of depression, PTSD, sleep disturbance, and pain offer specific targets for potential intervention that may improve veterans’ overall HRQOL perceptions. Indeed, these subjective measures of health status have long been identified as adversely afflicting Gulf War era veterans [9]. Finally, scientists have long been investigating potential root causes (e.g., oil well fires, pyridostigmine bromide pills, sarin gas, vaccines, infectious diseases, etc.) of GWI [63,64], but consensus has not yet been reached. For these reasons, we believe the association between GWI and each of the outcomes studied here are important for understanding the overall well-being of these veterans.

## 5. Conclusions

Veterans who met Kansas symptom criteria (regardless of exclusionary health conditions) and CDC GWI severe criteria reported worse physical and mental HRQOL scores, sleep disturbance, pain, and a higher proportion of reported clinical levels of PTSD and depression than veterans who did not meet the symptom criteria for the respective definitions. Importantly, Gulf War era veterans who may not meet strict definitions of GWI due to other diagnosable health conditions may experience the lowest HRQOL scores. As Gulf War veterans age, researchers and clinicians need to consider how aging-related health conditions and presentation of symptoms might modify their identification of GWI.

## Figures and Tables

**Figure 1 ijerph-19-04425-f001:**
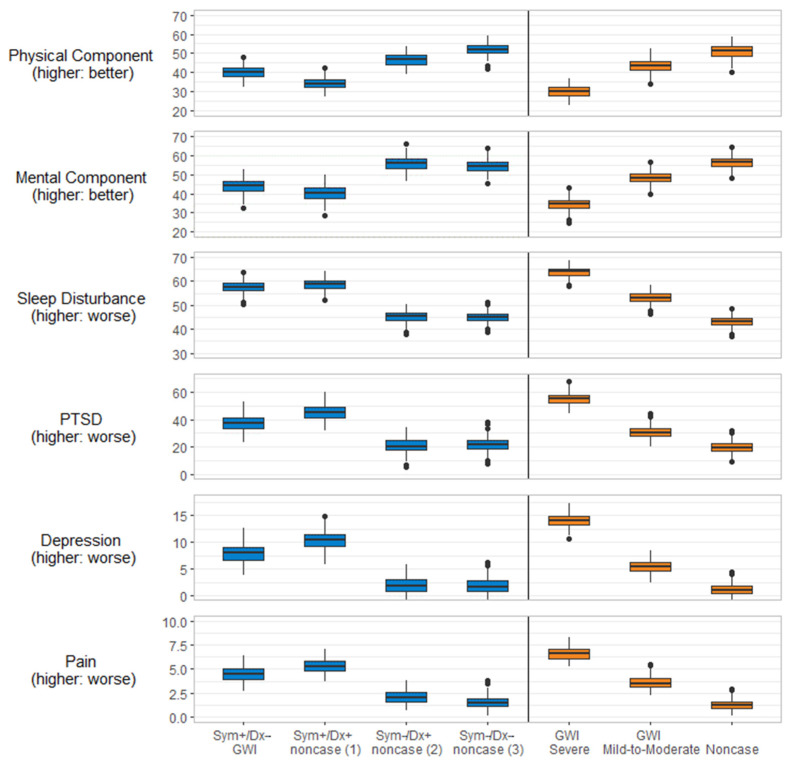
Adjusted boxplots of Health-related Quality of Life measures by Gulf War illness case status. Note: Model adjusted for age (continuous), sex, race/ethnicity, education, income, deployment status, military component, and military branch (see Appendix A for regression coefficients). Mean differences were compared across categories within the Kansas definition and alternatively across categories within the CDC definition. Comparisons were made to GWI for Kansas and for CDC GWI severe for CDC GWI categories. All differences were statistically significant (*p* < 0.001); sleep disturbance total score did not differ between KS GWI: Sym+/Dx− and KS noncase (1) Sym+/Dx+. GWI = Gulf War illness; PTSD = post-traumatic stress disorder; Sym = symptom criteria; Dx = Diagnoses.

**Figure 2 ijerph-19-04425-f002:**
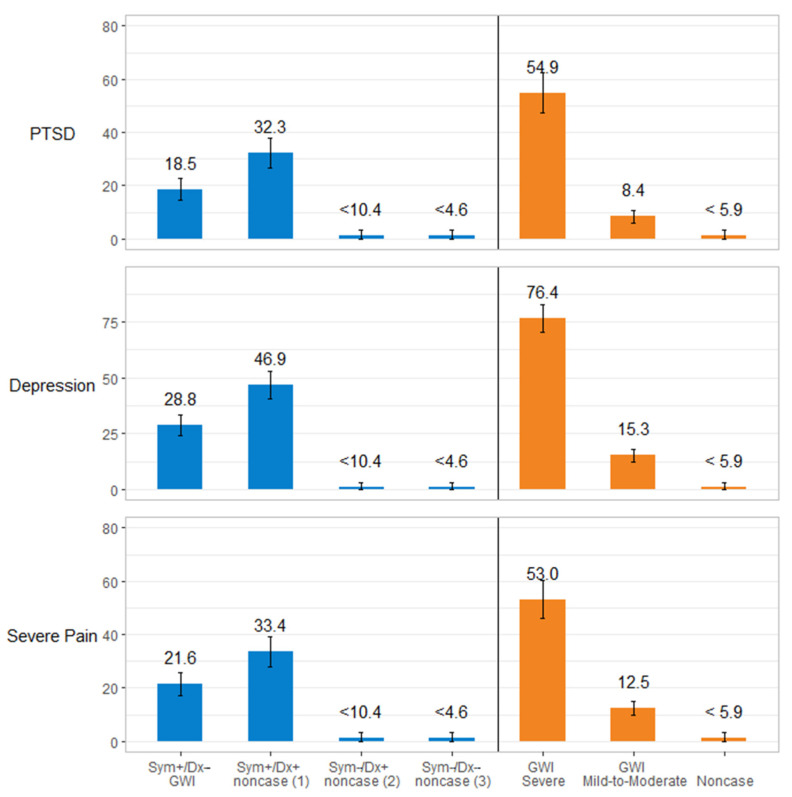
Proportion of GWECB Veterans meeting depression, post-traumatic stress disorder, and severe pain cutpoints by GWI Case Status, adjusted for demographic and characteristics of military service. Note: All values were suppressed due to fewer than ≤10) people in cell sizes for (a) KS noncase (2) Sym−/Dx+, (b) KS noncase (3) Sym−/Dx−, and (c) CDC noncases.

**Table 1 ijerph-19-04425-t001:** Descriptive Statistics by Gulf War Illness Case Status.

Characteristics	Total	Kansas				CDC		
		GWI Sym+/Dx−	Noncase (1) Sym+/Dx+	Noncase (2) Sym−/Dx+	Noncase (3) Sym−/Dx−	GWI Severe	GWI Mild-to-Moderate	Noncase
	(%)	(%)	(%)	(%)	(%)	(%)	(%)	(%)
Female	23.2	27.5	20.6 ^c^	21.7	20.1 ^c^	26.9	23.9	15.5 ^b^
Age								
<50 years	38.7	46.7	31.9 ^a^	24.5 ^a^	40.6	45.5	37.7 ^c^	32.6 ^b^
50–59 years	36.9	40.6	35.7	25.5 ^b^	37.2	34.8	38.3	34.8
60+ years	24.4	12.7	32.5 ^a^	50.0 ^a^	22.2 ^b^	19.7	23.9	32.6 ^b^
Race/ethnicity								
White, non-Hispanic	65.1	63.1	56.5	68.9	79.1 ^a^	45.5	70.2 ^a^	74.3 ^a^
Black, non-Hispanic	17.2	17.1	20.3	18.9	12.1	20.8	16.2	15.5
Hispanic	9.5	12.4	11.0	6.6 ^c^	Sup.^a^	18.6	7.1 ^a^	5.3 ^a^
Other	6.2	5.9	8.1	Sup.	5.0	11.7	4.7 ^a^	3.7 ^a^
Income								
<$30,000	11.2	12.4	14.2	Sup.	6.3 ^b^	16.7	9.6 ^b^	9.1 ^c^
$30,000-$59,999	23.1	22.8	31.0 ^c^	18.9	14.2 ^b^	29.9	22.7 ^c^	15.0 ^a^
$60,000-$99,999	29.0	27.5	29.0	34.0	29.7	28.4	30.1	26.2
$100,000+	29.7	31.5	17.7 ^a^	31.1	43.5 ^b^	16.7	31.1 ^a^	43.3 ^a^
Education								
<Associate’s degree	9.0	6.8	11.3 ^c^	14.2 ^c^	7.1	9.1	8.4	10.7
Bachelor’s or Associate’s degree	68.2	70.2	71.6	66.0	60.7 ^c^	75.8	67.1 ^b^	61.5 ^b^
Master’s, Professional, or Doctorate’s degree	20.7	20.7	13.9 ^c^	19.8	31.0 ^b^	12.1	22.3 ^a^	27.3 ^a^
Military Component								
Active Duty only	60.8	64.5	64.6	47.6 ^b^	54.6 ^c^	65.9	62.5	47.8 ^a^
Active Duty and Reserves	24.5	22.2	25.2	29.5	25.2	25.8	22.2	30.6
Reserves only	14.7	13.2	10.1	22.9 ^c^	20.2 ^c^	8.3	15.3 ^b^	21.5 ^a^
Military Branch								
Army	45.5	46.7	53.3	34.0 ^c^	37.2 ^c^	54.9	45.9 ^c^	31.0 ^a^
Navy	16.1	14.8	12.2	21.7	21.8 ^c^	11.7	15.5	24.6 ^a^
Airforce	11.0	11.0	9.3	15.1	11.7	7.2	12.0 ^c^	12.8
Marines	12.5	14.6	9.0 ^c^	12.3	14.2	11.7	12.2	15.0
National Guard	9.8	9.4	9.3	12.3	10.0	8.7	9.5	12.3
Other	5.0	3.5	7.0 ^c^	Sup.	5.0	5.7	5.0	Sup.
VHA user	44.6	47.6	61.7 ^a^	33.0 ^b^	20.1 ^a^	68.3	41.1 ^a^	24.2 ^a^
Deployed to Gulf	76.1	79.6	78.8	66.0 ^b^	70.3 ^b^	86.4	73.2 ^a^	71.7 ^a^
N	1116	426	345	106	239	264	665	187

Note: ^a^
*p* < 0.001, ^b^
*p* < 0.01, ^c^
*p* < 0.05 Note: For both the Kansas and CDC definitions, all chi-square analyses were statistically significant at *p* < 0.001 (i.e., Sex, age, race/ethnicity, income, education military unit, military branch, VHA user, and deployed). Not shown: missing or unknown race/ethnicity, unknown education, unknown income. Sup. = suppressed cell size < 11.

**Table 2 ijerph-19-04425-t002:** Descriptive Statistics: Veterans RAND-12, Sleep Disturbance, Post-Traumatic Stress, Depression, and Pain by GWI Case Status (unadjusted means).

	Total	Kansas				CDC		
		GWI Sym+/Dx−	Noncase (1) Sym+/Dx+	Noncase (2) Sym−/Dx+	Noncase (3) Sym−/Dx−	GWI Severe	GWI Mild-to-Moderate	Noncase
	Mean (SD)	Mean (SD)	Mean (SD)	Mean (SD)	Mean (SD)	Mean (SD)	Mean (SD)	Mean (SD)
Health-Related Quality of Life								
Physical Component	41.5	40.2	34.3 ^a^	46.9 ^a^	52.1 ^a^	30.0	43.5 ^a^	50.8 ^a^
	(12.3)	(11.3)	(11.3)	(10.5)	(6.5)	(10.1)	(10.7)	(8.0)
Mental Component	46.3	44.2	40.5 ^a^	56.0 ^a^	54.3 ^a^	34.6	48.1 ^a^	56.4 ^a^
	(12.7)	(12.5)	(12.8)	(7.2)	(7.5)	(11.1)	(11.1)	(6.5)
Other Indices of Health								
Sleep Disturbance	54.0	57.4	58.7	45.2 ^a^	45.0 ^a^	63.9	53.2 ^a^	43.1 ^a^
	(10.3)	(8.5)	(8.8)	(8.0)	(7.8)	(7.8)	(8.1)	(7.5)
PTSD	34.8	37.3	45.0 ^a^	20.7 ^a^	21.8 ^a^	55.3	30.8 ^a^	20.0 ^a^
	(17.3)	(15.9)	(18.2)	(4.7)	(6.7)	(15.1)	(12.6)	(6.1)
Depression	6.8	7.9	10.3 ^a^	1.9 ^a^	1.8 ^a^	14.1	5.5 ^a^	1.1 ^a^
	(6.2)	(5.8)	(6.3)	(2.2)	(2.4)	(5.4)	(4.5)	(2.1)
Pain	3.9	4.5	5.3 ^a^	2.0 ^a^	1.5 ^a^	6.6	3.5 ^a^	1.3 ^a^
	(2.8)	(2.6)	(2.7)	(2.2)	(1.7)	(2.3)	(2.4)	(1.9)

^a^*p* < 0.001. Mean differences were compared across categories within the Kansas definition and alternatively across categories within the CDC definition. Comparisons were made to KS GWI: Syx+/Dx− for Kansas and to CDC GWI severe for CDC GWI categories.

## Data Availability

GWECB data may be requested by following instructions on the GWECB repository website: https://www.research.va.gov/programs/csp/585/repository.cfm (accessed on 27 March 2022).

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
