# Peer review of "Health-Related Quality of Life by Gulf War Illness Case Status"

_ijerph, 2022, doi:10.3390/ijerph19084425_

Round 1

Reviewer 1 Report

I would like to thank the authors for writing their paper on the long-term health effects of having been deployed to the Gulf War. I think the authors have a nice sample of US veterans and interesting data on veterans’ quality of life and health status. Also interesting is that the data are gathered approximately 25 years after the Gulf War. The paper makes clear that veterans may experience long-term effects on their health and well-being of having been deployed to the Gulf War. That said, I would like to make several suggestions that may improve the paper.

Introduction

The paper aims to further our understanding of how veterans’ health related quality of life and other related indices differ across subtypes of Gulf War Illness case status, as assessed approximately 25 years after the war. In their introduction, the authors mention that heterogeneity within the Gulf War Illness case and control groups complicates our understanding of the extent to which Gulf War Illness is associated with veterans’ health-related quality of life. This is an interesting observation but it remains unclear if and how this problem will be addressed in the study. I think the authors should elaborate more on this heterogeneity in the introduction. What is heterogeneity? Why is heterogeneity a problem for our understanding of the extent to which Gulf War Illness is associated with veterans’ health-related quality of life? And, more importantly, how will this study help to solve that problem? In addition, the authors introduce two ways to assign Gulf War Illness case status to veterans. First by using the Kansas definition and second by using the definition as forwarded by the Centers for Disease Control and Prevention. What is missing in the introduction is what these definitions are - or what distinguishes one definition from the other - and it remains unclear why both definitions have to be included in the study. So, I think the authors should be more explicit in their introduction on what they see as the problem that they will address in their study, how their study will help to solve that problem, and their introduction should end with clearly stating the research question and/or their hypotheses that logically follow the problem that is outlined.

Method

I would welcome a more specific outline how their sample came about and who is in the sample. For example, they mention that a stratified sampling procedure was used but it remains unclear how the sample of respondents is related to the population. Related to that, the authors mention as a limitation that certain groups were underrepresented in the final sample, which may limit the generizability of findings to the broader population. However, as a reader one has no information on the population, nor a firm view on the sample, to find out what the implication of this limitation is.

I miss a clear overview of the percentages on the demographics and other (military service related) characteristics of the veterans in this study. They can be inferred from various tables, but I think they should be presented in a straightforward way.

Paragraph 3 and the appendix present an impressive amount of information based on various analyses. Following my comments regarding the introduction, I am not sure if all analyses and the resulting information are needed or that analyses that may be needed are missing. One thing that puzzles me is that two measures of Gulf War Illness were used in the study but I can’t find any information in the paper how these two measures are related to each other. I am also wondering what the effect of Gulf War Illness status is on health-related quality of life when controlled for the comorbid symptoms that are measured  (depression, PTSD, sleep disturbance, and pain).

Results

I think a more concise and consistent way of presenting the results will help the reader to more quickly grasp the essentials of it. So, I would suggest to choose a format (e.g. tables) and stick to that instead of presenting the information in tables, boxplots, graphs, using colors to indicate significance. Moreover, I would suggest to present the information on the variables that are used in the study in the same order as they were described/defined and not to vary the order in the text and tables, etc. (E.g., in the method section it is: depression, PTSD, sleep, pain; in table 1: sleep, PTSD, depression, pain, in figure 2: depression, PTSD, pain; in figure A2: pain, sleep, PTSD, depression). Also, for claritry reasons, I would suggest to use the names of the variables in the text and tables (e.g., depression, PTSD) instead of the instruments that were used to measure them. Additionally, I would suggest not to describe in the text what can be read from the tables, etc. but to highlight what is important, interesting, contrary to expectations, etc. Finally, I doubt whether it is necessary to refer to the instruments that were used to measure the variables in the notes to the tables, etc.

Discussion

In their discussion, the authors focus on the usefulness of the way veterans are assigned to one of the various Gulf War Illness case statuses. Also, in their limitations they mention that measures of Gulf War Illness and other indices of well-being and health in their study share common elements. Indeed, given the high correlations among Gulf War Illness-caseness and other symptoms that discussion on the usefulness is warranted. I guess this is related to the problem of heterogeneity that the authors mentioned in their introduction. Since heterogeneity within the Gulf War Illness case and control groups complicates our understanding of the extent to which Gulf War Illness is associated with veterans’ health-related quality of life, what has the study learned us to overcome the problem of heterogeneity?

What I find interesting is that Gulf War Illness status is assigned to veterans based on them reporting various symptoms and symptom severity but - as far as I can see - that they were not asked whether those symptoms were related to having been deployed to the Gulf War, 25 years earlier, as perceived by the veterans themselves. Besides the exclusionary illnesses that are mentioned when describing the Kansas definition, there may be various other reasons (that have not been told to you by a doctor or healthcare provider) that may evoke clusters of symptoms that may result in one being assigned to Gulf War Illness even if they are not related to having been deployed to the Gulf War - or any other mission - at all. Likewise, one can be excluded from having health or well-being related problems because of having been deployed to the Gulf War because one also has other health problems. Although attribution of symptoms itself may be problematic, I think not having asked whether a veteran attributes its symptoms to having been deployed is a limitation of the study that the authors should elaborate on in the discussion section.

Additional

I had some difficulty with reading the paper due to the abundant use of abbreviations and acronyms. What might help the reader, is to find some meaningful labels to distinguish between ‘Sym+/Dx-‘, ‘Sym-/Dx+’, ‘Sym+/Dx+’, ‘Sym-/Dx-‘, ‘CDC Severe’, ‘CDC Mild-to-Moderate’, ‘CDC Noncase’ and to use these labels throughout the paper.

In table 1, several values are suppressed. I think the authors should explain what this means; which they do elsewhere.

Author Response

Reviewer 1

Introduction

The paper aims to further our understanding of how veterans’ health related quality of life and other related indices differ across subtypes of Gulf War Illness case status, as assessed approximately 25 years after the war. In their introduction, the authors mention that heterogeneity within the Gulf War Illness case and control groups complicates our understanding of the extent to which Gulf War Illness is associated with veterans’ health-related quality of life. This is an interesting observation but it remains unclear if and how this problem will be addressed in the study. I think the authors should elaborate more on this heterogeneity in the introduction. What is heterogeneity? Why is heterogeneity a problem for our understanding of the extent to which Gulf War Illness is associated with veterans’ health-related quality of life? And, more importantly, how will this study help to solve that problem?

Response:  We have now edited the paragraph in the introduction to more clearly state the potential problems that heterogeneity in both symptom severity and use of exclusionary conditions presents in an aging cohort.

 In addition, the authors introduce two ways to assign Gulf War Illness case status to veterans. First by using the Kansas definition and second by using the definition as forwarded by the Centers for Disease Control and Prevention. What is missing in the introduction is what these definitions are - or what distinguishes one definition from the other - and it remains unclear why both definitions have to be included in the study.

Response:  The introduction now provides an overview of the definitions and highlights key differences in the two definitions.  The complete definition and how we operationalized the variables is described in the methods section

So, I think the authors should be more explicit in their introduction on what they see as the problem that they will address in their study, how their study will help to solve that problem, and their introduction should end with clearly stating the research question and/or their hypotheses that logically follow the problem that is outlined.

Response:  We now state a specific research question in the last paragraph of the introduction: “How do HRQOL and indices of health outcomes vary across subgroups of veterans based on GWI case status?”

I would welcome a more specific outline how their sample came about and who is in the sample. For example, they mention that a stratified sampling procedure was used but it remains unclear how the sample of respondents is related to the population. Related to that, the authors mention as a limitation that certain groups were underrepresented in the final sample, which may limit the generizability of findings to the broader population. However, as a reader one has no information on the population, nor a firm view on the sample, to find out what the implication of this limitation is.

Response:  We have added additional information to the data section regarding the sample design which is more extensively descried here: Khalil, L. et. al. The Gulf War Era Cohort and Biorepository: A longitudinal research resource of veterans of the 1990–1991 Gulf War Era. Am. J. Epidemiol. 2018, 187, (11), 2279-2291, 10.1093/aje/kwy147.

I miss a clear overview of the percentages on the demographics and other (military service related) characteristics of the veterans in this study. They can be inferred from various tables, but I think they should be presented in a straightforward way.

Response: We have added a total column to table 1 and a brief description of the sample in the results section.

Paragraph 3 and the appendix present an impressive amount of information based on various analyses. Following my comments regarding the introduction, I am not sure if all analyses and the resulting information are needed or that analyses that may be needed are missing. One thing that puzzles me is that two measures of Gulf War Illness were used in the study but I can’t find any information in the paper how these two measures are related to each other. I am also wondering what the effect of Gulf War Illness status is on health-related quality of life when controlled for the comorbid symptoms that are measured  (depression, PTSD, sleep disturbance, and pain).

Response  The purpose of this paper is to document differences between veterans by Gulf War Illness status.  We have integrated information into the strengths and limitations section about why it is so important to document rates of HRQOL and indices of health across the subtypes of GWI.  We believe that our analysis documents important disparities across GWI subtypes. The full list of symptoms for the Kansas and CDC GWI definitions are included in Tables A1 and A2.  We also reference a previous paper that describes in even more detail how the definitions were operationalized in our cohort: Gifford, E. J.; Vahey, J.; Hauser, E. R.; Sims, K.; Efird, J. T.; Dursa, E. K.; Steele, L.; Helmer, D. A.; Provenzale, D., Gulf War illness in the Gulf War Era Cohort and Biorepository: The Kansas and centers for disease control definitions. Life Sci 2021, 119454, 10.1016/j.lfs.2021.119454.

Results

I think a more concise and consistent way of presenting the results will help the reader to more quickly grasp the essentials of it. So, I would suggest to choose a format (e.g. tables) and stick to that instead of presenting the information in tables, boxplots, graphs, using colors to indicate significance. Moreover, I would suggest to present the information on the variables that are used in the study in the same order as they were described/defined and not to vary the order in the text and tables, etc. (E.g., in the method section it is: depression, PTSD, sleep, pain; in table 1: sleep, PTSD, depression, pain, in figure 2: depression, PTSD, pain; in figure A2: pain, sleep, PTSD, depression).

Response: We have re-ordered the presentation of the variables and results for consistency.  We have updated the figures to reduce the number of colors and improve the labeling of the information in the figures. Table 2 presents the unadjusted means and Figure 1 presents the adjusted means.  While we like the precision of information that is displayed in the table, we do believe that the figure offers the reader quick insight into the relationship of GWI case status and the measures that is much more difficult to discern from the tables.  We do provide a table in the appendix (Table A6) with the information contained in Figure 2.

Also, for clarity reasons, I would suggest to use the names of the variables in the text and tables (e.g., depression, PTSD) instead of the instruments that were used to measure them. Additionally, I would suggest not to describe in the text what can be read from the tables, etc. but to highlight what is important, interesting, contrary to expectations, etc. Finally, I doubt whether it is necessary to refer to the instruments that were used to measure the variables in the notes to the tables, etc.

Response:  We have removed notes in tables and have tried to refer to the measures by the construct they are measuring.  We have edited the text and tables to more consistently  refer to  the construct being measured rather than instrument used. 

In their discussion, the authors focus on the usefulness of the way veterans are assigned to one of the various Gulf War Illness case statuses. Also, in their limitations they mention that measures of Gulf War Illness and other indices of well-being and health in their study share common elements. Indeed, given the high correlations among Gulf War Illness-caseness and other symptoms that discussion on the usefulness is warranted. I guess this is related to the problem of heterogeneity that the authors mentioned in their introduction. Since heterogeneity within the Gulf War Illness case and control groups complicates our understanding of the extent to which Gulf War Illness is associated with veterans’ health-related quality of life, what has the study learned us to overcome the problem of heterogeneity?

Response:  We have now better defined in the introduction heterogeneity as arising from two sources—the first is how exclusionary health conditions are applied and the second is severity.  In the strengths and limitations section we now describe how researcher should consideration these factors in future work on the aging process.  This is essential because the cohort is aging and addressing the needs of these veterans will require an understanding of GWI within the context of aging.

What I find interesting is that Gulf War Illness status is assigned to veterans based on them reporting various symptoms and symptom severity but - as far as I can see - that they were not asked whether those symptoms were related to having been deployed to the Gulf War, 25 years earlier, as perceived by the veterans themselves. Besides the exclusionary illnesses that are mentioned when describing the Kansas definition, there may be various other reasons (that have not been told to you by a doctor or healthcare provider) that may evoke clusters of symptoms that may result in one being assigned to Gulf War Illness even if they are not related to having been deployed to the Gulf War - or any other mission - at all. Likewise, one can be excluded from having health or well-being related problems because of having been deployed to the Gulf War because one also has other health problems. Although attribution of symptoms itself may be problematic, I think not having asked whether a veteran attributes its symptoms to having been deployed is a limitation of the study that the authors should elaborate on in the discussion section.

Response: The limitations section now describes the relationship of  symptom attribution to veteran outcomes and cites previous studies on this matter.

I had some difficulty with reading the paper due to the abundant use of abbreviations and acronyms. What might help the reader, is to find some meaningful labels to distinguish between ‘Sym+/Dx-‘, ‘Sym-/Dx+’, ‘Sym+/Dx+’, ‘Sym-/Dx-‘, ‘CDC Severe’, ‘CDC Mild-to-Moderate’, ‘CDC Noncase’ and to use these labels throughout the paper.

Response :  We agree that the number of groups can burden the reader.  We added a prefix to the KS categories which hopefully are informative and clear “KS GWI: Sym+/Dx-“,  “KS noncase(1): Sym-/Dx+”, “KS noncase(2): Sym+/Dx+”,KS noncase(3):  Sym-/Dx-“.  Similar, we clarified the CDC categories so that it should be easier for the reader to immediately see that 2 of the 3 categories may be considered GWI: CDC GWI Severe”, “CDC GWI Mild-to-Moderate”,CDC Noncase”.

In table 1, several values are suppressed. I think the authors should explain what this means; which they do elsewhere.

Response: This information has now been added to table 1.

Reviewer 2 Report

Dear author,

First of all, congratulations for the work done and the effort involved in conducting research of this nature.

It provides a good conceptualisation, design, good analysis and discussions along with limitations, social implications.

Congratulations on the work done.

However, I would suggest a number of recommendations:

-In the introduction justify with better article the relations of quality of life with sleep.

-Incorporate what theoretical implications this work has to the scientific community.

-Incorporate what practical implications this work has for the community/society/population.

-To incorporate social implications.

-Incorporate strengths of your work.

Best regards

Author Response

-In the introduction justify with better article the relations of quality of life with sleep.

Response: We have added two additional citations on the relationship between sleep and quality of life:

Verster, J. C. P.-P., S.R.; Streiner, D.L. , Sleep and Quality of Life in Clinical Medicine. Springer: Berlin/Heidelberg, Germany,, 2008.

Matsui, K.; Yoshiike, T.; Nagao, K.; Utsumi, T.; Tsuru, A.; Otsuki, R.; Ayabe, N.; Hazumi, M.; Suzuki, M.; Saitoh, K.; Aritake-Okada, S.; Inoue, Y.; Kuriyama, K., Association of subjective quality and quantity of sleep with suality of life among a general population. Int J Environ Res Public Health 2021, 18, (23), 10.3390/ijerph182312835.

-Incorporate what theoretical implications this work has to the scientific community.

-Incorporate what practical implications this work has for the community/society/population.

-To incorporate social implications.

-Incorporate strengths of your work.

Response:  We have now created a strengths and limitation section in which we document the implications of this study’s findings to the audiences mentioned above.

Reviewer 3 Report

This manuscript regarding different definition of Gulf War Illness (GWI) and its association with adverse health outcome and quality of life was carefully designed and well written. I only have few comments:

  1. From definition of GWI, whether Kansas or CDC definition, participants with GWI may have developed cognitive impairment, even severe cognitive impairment. Moreover, certain outcome measurement of this study including sleep disturbance and depression were related to cognitive impairment. Due to all variables were subjectively recorded, answers from participants with severe cognitive impairment may biased the results of the study. Please put some words regarding how you avoid this kind of bias.
  2. Multimorbidity has been investigated among older adults and different multimorbidity patterns are related to mortality, disability and other adverse outcomes. From your results, participants with (Sym +/ Dx +) could have been diagnosed of multiple comorbidities. For example, participants with both cancer and diabetes may experience worse outcomes than participants with only cancer or diabetes. Or, participants with hepatitis C may experience no difference with participants with (Sym +/ Dx -). I think it will give readers more information when subgroup analysis regarding different comorbidity is performed.  
  3. Other than comorbidities, disability have been proved to be related to QoL. Participants of disability may experienced more depressed mood, sleep problem or poorer QoL. Have you considered the impact of disability on outcome measurements among participants? 
  4. Please perform spell check and correct typos in the manuscript. 

Author Response

From definition of GWI, whether Kansas or CDC definition, participants with GWI may have developed cognitive impairment, even severe cognitive impairment. Moreover, certain outcome measurement of this study including sleep disturbance and depression were related to cognitive impairment. Due to all variables were subjectively recorded, answers from participants with severe cognitive impairment may biased the results of the study. Please put some words regarding how you avoid this kind of bias.

Response:  We now address this in the limitations section.

Multimorbidity has been investigated among older adults and different multimorbidity patterns are related to mortality, disability and other adverse outcomes. From your results, participants with (Sym +/ Dx +) could have been diagnosed of multiple comorbidities. For example, participants with both cancer and diabetes may experience worse outcomes than participants with only cancer or diabetes. Or, participants with hepatitis C may experience no difference with participants with (Sym +/ Dx -). I think it will give readers more information when subgroup analysis regarding different comorbidity is performed.  

Response:  This is true that multimorbidity may be associated with additional health concerns. We have provided 7 subgroups and suggest that future work might consider additional subdivisions of comorbidity. Further, in this cohort, we would start to face sample size limitations if we further divide the sample by comorbid groups. However, future analyses of other larger cohorts (e.g., the Million Veterans Program) may incorporate this approach.  In our strengths and limitations section, we now describe a strength of the study as documenting the association between GWI and HRQOL and other health indices.  Future work with this cohort is planned to do an in-depth examination of the accumulation of comorbidities in the aging process.  We believe that this work supports the need for such an endeavor.

Other than comorbidities, disability have been proved to be related to QoL. Participants of disability may experienced more depressed mood, sleep problem or poorer QoL. Have you considered the impact of disability on outcome measurements among participants?  

Response:  We have added a sentence to the limitations section to note that disabilities may be associated with our measures and that we were unable to study them here.

Please perform spell check and correct typos in the manuscript. 

Response:  We edited the manuscript for spelling and grammar.

Round 2

Reviewer 1 Report

I would like to thank the authors for addressing my suggestions.

Author Response

Reviewer Comment 1: I think the authors should follow the reviewers' comments and report a crosstabulation between the two definitions of GWI.  This I think would be essential.

Response: We have now included an appendix table that demonstrates the crosstabulation between the two definitions of GWI.

Reviewer Comment 2: The authors should recognize earlier research that looked at subjective health status as a function of GWI.

Response: We have now included a citation to this article which examined subjective health status between Gulf War era veterans with veterans from the Bosnia war and other deployments.

Hotopf, M., David, A. S., Hull, L., Nikalaou, V., Unwin, C., & Wessely, S., Gulf war illness—better, worse, or just the same? A cohort study. BMJ 2003, 32, (7428), 1370, doi: https://doi.org/10.1136/bmj.327.7428.1370.

Reviewer Comment 3: Controlling for deployment status is important because some non-deploying military personnel may have received some of the same exposures (vaccines, PB pills) that deployed personnel did.  

Response:  We agree that controlling for deployment is important. The models control for deployment.  This is described in the methods section and the coefficient for deployment is shown in appendix tables A4 and A5.

Reviewer Comment 4: The potential role of PB pills and vaccines should be acknowledged (see RAND series on GWI).

Response:  We now reference the RAND study and the following Institute of Medicine Report.

RAND Examining possible causes of Gulf War illness; RAND National Defense Research Institute: Santa Monica, California.

Institute of Medicine, Depleted uranium, pyridostigmine bromide, sarin, and vaccines. National Academy Press: Washington, D.C., 2000; Vol. 1.